# Association between whole grain food intake in Canada and nutrient intake, food group intake and diet quality: Findings from the 2015 Canadian Community Health Survey

Jessica Smith[1]*, Yong Zhu[1], Neha Jain[2], Norton Holschuh[3]

1 Bell Institute of Health and Nutrition, General Mills, Minneapolis, Minnesota, United States of America,
2 Global Knowledge Solutions, General Mills, Mississauga, Ontario, Canada, 3 Global Knowledge Solutions,
General Mills, Minneapolis, Minnesota, United States of America

* jessica.smith@genmills.com

pone.0253052

HONG KONG

**Data Availability Statement:** Canadian Community
Health Survey - Nutrition 2015 https://www150.
statcan.gc.ca/n1/en/catalogue/82M0024X2018001.

## Abstract

Whole grains have been associated with numerous beneficial health outcomes and are rec-
ommended in Canada's Food Guide; however, there is little research on whole grains spe-
cific to Canada. Therefore, the objective of this study was to characterize the association
between Canadians' WG intake and nutrients, food groups and diet quality and to under-
stand top sources of WG in the diets of Canadians. We used data from the Canadian Com-
munity Health Survey 2015: a cross-sectional survey that collected information on diet
(using a 24-hour recall) and health from 20,487 Canadians 1 year and older. We classified
study participants according to their WG intake: non-WG (n = 10,883) and three groups
based on age-specific tertiles of WG intake, low-WG (n = 3,322), mid-WG (n = 3,180), and
high-WG (n = 3,102). Results were analyzed using population-based survey methods and
were adjusted for energy, age, gender, overweight/obesity, income, and supplement use.
We found differences in nutrients and food groups by WG group: there was a significant lin-
ear trend across groups of increasing WG for increased fiber (children and adults), vitamin
B$_6$ (children), thiamin (adults), potassium (children and adults), zinc (adults), calcium (chil-
dren and adults), iron (children and adults), magnesium (children and adults), fruit (adults),
and legumes, nuts and seeds (adults); and decreased total fat (adults), saturated fat
(adults), folate (children and adults), refined grains (adults and children), and meat and poul-
try (adults) intake. We found that there were no differences in total sugar or sodium intake
across WG intake groups. The high WG intake group for both children and adults had higher
diet quality, measured by the Nutrient Rich Food Index 9.3, compared to non-WG eaters.
The top 2 food sources of WG across WG intake groups for children and adults were whole
grain oat and high fiber breakfast cereal and whole grain and whole wheat bread. Other top
sources of WG included rice, bread products, other breakfast cereals, salty snacks, cereal
grains and flours, pasta, and sweet snacks. This research supports recommendations to
increase WG foods intake as a means to improve diet quality of Canadians.

**Funding:** This study was funded by the Bell Institute of Health and Nutrition, General Mills. All authors are employed by General Mills, a food manufacturer of grain-based foods, among other food categories, and were responsible for the study design, analysis, decision to publish and the preparation of the manuscript. The data analyzed in this study are from a nationally representative survey that are publicly available and the funder played no role in the original survey design or survey methodology.

**Competing interests:** All authors have the following competing interests: all authors (JS, YZ, NJ, NH) are employees of General Mills, Inc., a food manufacturer. This conflict of interest does not alter our adherence to PLOS ONE policies on sharing data and materials.

## Introduction

Canada's Dietary Guidelines recommend consumption of whole grain foods as part of healthy dietary patterns [1]. Numerous observational data support the association between whole grain intake and positive health outcomes including decreased risk and mortality from cardiovascular disease, type 2 diabetes, and cancer [2–6]. Whole grain intake may have a direct biological impact on the etiology of these diseases, potentially via influence on the microbiome [7,8] or other pathways [9]. Additionally, consumption of whole grains may be a marker of higher overall diet quality and be associated with beneficial intake of numerous food groups, nutrients, and dietary compounds [6,10,11]. Despite these benefits, intakes of whole grain remain below recommendations in the US and Europe [12–16]. Few global policies have been directed toward increasing whole grain intake despite potentially large impacts on decreasing mortality and disability [17].

Canada's Healthy Eating Strategy is a multi-pronged approach to "improve the food environment in Canada to make it easier for Canadians to make the healthier choice [18]." Proposed and implemented policies include changes to the nutrition facts panel, development of the new Canada Food Guide, proposed mandatory front-of-pack labeling, proposed mandatory food marketing restrictions, and voluntary food-category specific sodium targets. The goals of the strategy include improving healthy eating information; improving nutrition quality of foods; protecting vulnerable populations; and supporting increased access to and availability of nutritious foods. However, the majority of these policies are focused on reducing intake of specific nutrients, namely sodium, saturated fat and sugar, while little attention has been paid to encouraging increased consumption of beneficial food groups, such as whole grains, particularly among Canadians with lower quality diets, such as individuals living with food insecurity [18].

Policy efforts in Canada may be hindered by a lack of information on current whole grain intake and its association with diet quality in Canada. A modeling study using Canadian data found that the majority of the economic burden of chronic diseases attributable to unhealthy eating was due to low consumption of nuts and seeds and whole grain [19], highlighting the importance and relevance of understanding the association between whole grain intake and intake of other recommended food groups and nutrients. Therefore, the objective of this study was to characterize the dietary intakes of Canadians according to their intake of whole grain foods and understand top sources of whole grains in the diets of Canadians. These findings can be used by policy makers, educators, and health care providers to encourage consumption of whole grains among Canadians.

## Materials and methods

### Study design and population

This study used the publicly available data from the 2015 Canadian Community Health Survey (CCHS)–Nutrition: Public Use Microdata File (PUMF), which can be accessed through Statistics Canada [20]. CCHS 2015 is a cross-sectional survey that collected information on diet and health from Canadians 1 year of age and older living in private dwellings in the ten provinces of Canada in 2015. The study was administered by Statistics Canada and used a multi-stage clustered design to generate a representative sample (based on urban or rural residence, age, gender, and socioeconomic status) of Canadians that covered 90% of the population in the ten provinces. Using their multistage cluster sampling design, a single respondent from each household was selected to participant in the study. The study targeted a sample size of 24,000 and ultimately collected food intake data on 20,487 respondents [21].

Dietary information was collected using a 24-hr dietary recall conducted using the Automated Multiple Pass Method which is an in-person computer-assisted interview, originally developed by the United States Department of Agriculture (USDA). Among a subset of CCHS 2015 participants (n = 7,608), a second 24-hr dietary recall was conducted in order to assess the within-person variation in an individual's nutrient intake data. Because only a subset of participants had an available second 24 hr dietary recall, we used only the first dietary recall for all study participants. Proxies (parent or guardian) were used to report the dietary intake for children 1 to <6 years. Children 6 to 11 years were assisted in completing the dietary recall by a parent or guardian and children over 12 years of age completed the survey on their own. We included all participants with a 24 hr recall in the current study. Very few participants had extreme energy intakes <200 kcal/day (n = 3 for children; n = 36 for adults) or >8000 kcal/day (n = 1 for children; n = 3 for adults). Health information was also collected via physical exam and questionnaire and included information on supplement use, general health, physical activity, sociodemographic characteristics, and food security. Appropriate ethics approval and informed consent from participants was obtained by the CCHS 2015 study. The PUMF data are developed from the complete CCHS 2015 data to ensure respondent confidentiality with "each PUMP [going] through a formal review and approval process by an executive committee of Statistics Canada."

## Whole grain intake

Whole grain intake was defined according to consumption of whole grain foods on the first 24-hr dietary recall. Foods from the 24-hr recalls were coded into groups according to the 2007 Canada Food Guide (CFG) food groups and subgroups. CFG food groups are classified into 4 tiers based on how closely they align with the 2007 CFG, with tiers 1 and 2 aligned with the food guide, tier 3 partially aligned, and tier 4 not aligned. Not all foods were mapped to CFG food groups– 49% of food codes (primarily mixed dishes) were not assigned to a CFG group.

Whole grain foods were defined according to the CFG food group "Grain products–whole" tiers 1 through 4. We included all tiers of whole grain foods to capture whole grain intake as completely as possible, while adhering to foods that were assigned to the CFG food groups. Examples of tier 1 whole grain foods include plain oatmeal, bulgur, whole wheat flour and brown rice; tier 2 foods include whole wheat bread, whole wheat crackers, and ready-to-eat multigrain cereal; tier 3 included low-fat microwaved popcorn, whole wheat dried fruit and nut cookies, whole wheat biscuits, and granola; lastly, tier 4 products included homemade granola, buttered popcorn and a whole wheat fruit and nut cake. We only considered whole grain intake from foods classified as whole grain according to the CFG food groups, and not whole grain from mixed dishes, which were not classified into CFG food groups. S1 Table includes a list of individual food codes from mixed dishes that included the terms "whole grain" "whole wheat" or "wholemeal" in their description.

Study participants were classified according to their consumption of grams of whole grain food intake. We also reported the average number of servings of whole grain foods, as defined in the CCHS 2015 database, and the percent of grain foods intake from whole grain foods, calculated as the mean of the ratios of grams of total grain food tiers 1–4 (defined as refined grains tier 1–3 + refined grains tier 4 + whole grains tier 1–3 + whole grains tier 4) to whole grain food tiers 1–4. Participants that did not report consuming any foods classified as whole grain were categorized as no-WG consumers. Participants that did report consumption of at least one whole grain food were divided according to age-specific tertiles of grams of whole grain foods consumed. Four groups were created: non-whole grain food eaters (no-WG; 1 to 18

years n = 3305; 19 years and older n = 7578); low-whole grain food eaters (low-WG; 1 to 18 years n = 1077; 19 years and older n = 2044), mid-whole grain food eaters (mid-WG; 1 to 18 years n = 1085; 19 years and older n = 2156); and high-whole grain food eaters (high-WG; 1 to 18 years n = 1101; 19 years and older n = 2141).

## Nutrient intake

Average nutrient intake from the 1$^{st}$ 24 hr recall were reported according to the 4 groups of whole grain intake. The Canadian Standard Food Composition database, a modified version of the United States Department of Agriculture (USDA) recipe database, was used to calculate nutrient intakes from the foods reported consumed in the 1$^{st}$ 24 hr recall. Quantitative amounts of nutrients obtained from supplements were not reported in the CCHS public use microfile data and, therefore, results in the present analysis represent nutrient intake from diet only.

## Food group intake

Canada Food Guide (CFG) food groups were used to define food group categories for this study. There were 30 food groups with four tiers for each food group identified in CCHS. The four tiers indicate whether a food item is aligned, partially aligned or not aligned with the 2007 CFG [22]. We grouped the 30 food groups in CCHS into 12 categories for our study and further grouped foods in tiers 1–3 together (combining the aligned or partially aligned foods) and tiers 4 together (foods not aligned with CFG). See S2 Table for details on food group definitions. The CFG food groups categorized individual food codes according to their food group and their nutritional composition (i.e. their tiers). We relied on the CFG food groups to provide insights into how whole grain intake was associated with recommended food groups within the Canadian Food Guide. This information is complementary to the Nutrient Rich Food Index (NRF), used to measure dietary quality.

## Top sources of whole grain foods

Top sources of whole grain foods were based on the average grams of intake of foods classified according to the BNS food groups in CCHS. According to the study documentation: "The 'BNS food and recipe groups' were developed by the Bureau of Nutritional Sciences (BNS) at Health Canada in the early 1990s based on the British and American food group systems [20]." We used the BNS food groups to report the top sources of whole grain in order to provide more granularity on the types of foods that were consumed within the CFG whole grain tier 1–4 food group. BNS food groups categorize individual food codes according to similarities in their dietary usage and nutritional content and "provides the means a) to categorize and summarize the detailed food and recipe information collected in nutrition surveys and b) to facilitate analyses of the composition of the diet [20]." Unlike the CFG food groups, we used the BNS food groups to report the top sources of WG in the diet of Canadians since the BNS food groups provide further granularity into the types of foods consumed rather than their broad food group or alignment with the Canada Food Guide.

## Diet quality

Diet quality was assessed using a modified version of the Nutrient Rich Food Index (NRF) 9.3 [23]. Because CCHS data lacks quantitative information on total intake of food groups from all foods (i.e. mixed dishes are not disaggregated), we selected an index of dietary quality that relied on nutrients only. Several previous studies have used NRF 9.3 as a measure of diet

quality [24–27]. NRF 9.3 was developed to measure the nutrient density of individual foods by calculating the sum of 9 nutrients to encourage calculated as a percentage of daily recommendations (for a 2000 kcal diet; capped at 100% to avoid skewing scores based on very high intakes of a single nutrient) minus the sum of 3 nutrients to limits (calculated as a percentage of daily recommendations but with no cap to allow for high intake of these nutrients to reduce the diet quality score) and has been reported to positively correlate (in both children and adults) with the Healthy Eating Index, an established measure of diet quality [23]. The 9 nutrients to encourage were fiber, protein, vitamin D, vitamin C, iron, calcium, potassium, vitamin A, and magnesium and the 3 nutrients to limit were total sugar, sodium, and saturated fat. Added sugar intake was not reported in CCHS data, therefore, total sugar was used in the NRF calculation [23]. Daily values (DV), developed by Health Canada, were used as daily recommendations and can be found in S3 Table [28]. The NRF 9.3 was modified for the total diet by calculating the percent of the DV for each nutrient per 2000 kcal of the total diet for each individual [24,29]. A higher NRF 9.3 score indicates higher diet quality with a theoretical maximum possible score of 900 (100% for all nutrients to encourage and 0% for all nutrients to limit per 1000 kcal).

## Covariates

We included the following covariates, selected a priori based on their plausible associations with our outcomes of interest, in our model as potential confounding variables in the relationship between whole grain intake and dietary quality: total energy intake (categorical), age (continuous), gender (categorical), overweight/obesity status (yes/no), low-income (categorical), and supplement use (categorical). Total energy intake was included as a categorical variable to account for the non-linear trend in energy intake that was see across groups of WG intake. Participants were classified as overweight/obese according BMI $\geq$25 kg/m$^2$ for adults or BMI z-scores for children $\geq$75$^{th}$ percentile. Low-income was defined using total household size and total household income using similar thresholds to those in the Canadian 2016 Census of Population Dictionary [30]. For households with 1 person, low-income was defined as a total household income from all sources $\leq$$19,999CAD; for households of 2 or 3 persons, low-income was $\leq$$39,999CAD; for households of 4 and 5 or more persons, the threshold was $\leq$$59,999CAD. We found no evidence of collinearity between our covariates in our model (data not shown).

## Statistical methods

All analyses were done using SAS version 9.4 (Cary, NC, United States) using survey methodologies incorporating appropriate study weights, as provided by the CCHS 2015 study documentation. Demographic data are presented as unadjusted means and standard errors or percentages, as indicated in the results. All other results are presented as least squared means, adjusted for the previously mentioned covariates using a multivariable linear regression model. We assumed that each participant was independent of each other, as only a single respondent was selected from each household. Statistical significance was assessed using linear trends across the ordinal WG intake groups for continuous variables or Chi$^2$ for categorical variables and we applied a Bonferroni correction for multiple comparisons for each outcome separately for each of our populations (i.e. children and adults). A p$\leq$0.002 was considered statistically significant for both nutrient intakes (0.05/22 nutrients = 0.00227) and for food group intakes (0.05/24 food groups = 0.00208). A p value <0.05 considered significant for demographic characteristics, as these were descriptive results and not part of the study objective, and for diet quality data, because there was only a single value for this outcome.

## Results

### Demographic characteristics

Half of children and adolescents (50%) and slightly more adults (54%) were classified as no-WG consumers. Distribution across the three WG intake groups was evenly distributed (by design according to age-specific tertiles of WG intake) with 16%, 17% and 17% of children being classified as low-, mid- and high- consumers, respectively. For adults, 15% were classified each as low-, mid- and high-WG consumers. Whole grain food intake was 0 g/d, and contributed to 0% of total grain intake, for children and adults in the no-WG group, by design. For the low-, mid- and high-WG intake groups, whole grain food intake ranged from 19.3 g/d to 163.1 g/d (or 0.6 to 3.4 servings/day) for children and 28.9 g/d and 219.9 g/d (or 0.9 to 3.7 servings/day) for adults and the percent of total grain food intake from whole grain foods ranged from 9% to 49% for children and from 15% to 61% for adults. For children, but not adults, unadjusted energy intake differed significantly across groups of whole grain intake with the highest energy intake found among the high-WG group followed by the no-WG group. Among children, the low-WG group had the lowest calorie intake of all 4 groups (Table 1).

Amongst the demographic data, we found differences for both children and adults between the whole grain intake groups in age, gender, smoking status, and supplement use according to level of whole grain intake in the total Canadian population 1 year and older (Table 1). For children only, there were differences in official language and meeting physical activity guidelines (Table 1). Individuals with the highest intake of whole grains foods (high-WG) were older (children 10.0 ± 0.2 years; adults 50.5 ± 0.6 years) compared to no-WG for adults (47.7 ± 0.4 years, p<0.0001 linear trend across all 4 groups of whole grain intake) or low-WG for children (7.3 ± 0.3 years, p<0.0001 linear trend). The percent of girls and women also varied across intake of whole grain such that most were in the low-WG group (children 55.2%; adults 63.6%) and the least fell in the high-WG group (children 40.6%; adults 44.1%, p<0.0001 across all 4 groups of whole grain intake for both children and adults).

Participants in the no-WG group were the most likely to be daily or occasional smokers (children 2.1%; adults 22.3%) whereas participants in the high-WG group were the least likely (children 1.1%; adults 10.6%; p<0.0001 for children and p = 0.0002 for adults across all 4 groups of whole grain intake). Participants in the no-WG group were the least likely to be taking a nutritional supplement (children 34.5% yes; adults 40.7% yes) and participants in the high-WG group for adults (56.8% yes, p<0.0001 across all 4 groups of whole grain intake) and the mid-WG group for children (44.7% yes, p = 0.004 for trend) were the most likely to be taking a nutritional supplement. All other demographic characteristics, including education, marital status, prevalence of overweight/obesity, low-income status, immigration status, and food insecurity did not differ.

### Nutrient intakes

Daily nutrient intakes by whole grain consumption and age group is presented in Table 2. After adjusting for energy intake and other potentially confounding variables, we saw significant differences in nutrient intake across groups of whole grain intake for both children and adults (Table 2, unadjusted results in S4 Table). Total carbohydrate intake increased (both absolute grams and as a percent of energy) with increasing whole grain intake but total sugar intake did not differ for both children and adults. Protein intake was not different, in grams or as a percent of energy, across groups. We observed large differences in fiber intake between the no-WG group and the high-WG group. Specifically, children in the high-WG group had a 53% higher fiber intake compared to the no-WG group and for adults this difference was 63%.

**Table 1. Demographic characteristics for Canadian children and adults stratified by whole grain food intake.**

| | | No Whole Grain Food Intake (No-WG)[a] | | Low Whole Grain Food Intake (Low-WG)[a] | | Mid-Whole Grain Food Intake (Mid-WG)[a] | | High-Whole Grain Food Intake (High-WG)[a] | | p value[b] | |
|---|---|---|---|---|---|---|---|---|---|---|---|
| | | Children n = 3,305 | Adults n = 7,578 | Children n = 1,077 | Adults n = 2,044 | Children n = 1,085 | Adults n = 2,156 | Children n = 1,101 | Adults n = 2,141 | Children | Adults |
| Age, years mean ± SE | | 10.2 ± 0.2 | 47.7 ± 0.4 | 7.3 ± 0.3 | 50.8 ± 1.0 | 9.0 ± 0.3 | 50.8 ± 1.1 | 10.0 ± 0.2 | 50.5 ± 0.6 | <0.0001 | <0.0001 |
| Whole grain food intake, g/d mean ± SE | | 0 ± 0 | 0 ± 0 | 19.3 ± 0.9 | 28.9 ± 0.6 | 54.5 ± 1.3 | 71.6 ± 0.8 | 163.1 ± 5.7 | 219.9 ± 12.1 | <0.0001 | <0.0001 |
| Whole grain food intake, number of servings/day mean[c] ± SE | | 0.0 ± 0.0 | 0.0 ± 0.0 | 0.6 ± 0.0 | 0.9 ± 0.0 | 1.6 ± 0.0 | 2.1 ± 0.0 | 3.4 ± 0.1 | 3.7 ± 0.3 | <0.0001 | <0.0001 |
| Percent of total grain food intake from whole grain foods[d], % | | 0 ± 0% | 0 ± 0% | 15.7 ± 0.8% | 26.3 ± 1.1% | 32.6 ± 1.4% | 45.3 ± 2.2% | 54.0 ± 1.3% | 65.3 ± 1.1% | <0.0001 | <0.0001 |
| Energy intake, kcal | | 1809 ± 23 | 1878 ± 30 | 1604 ± 39 | 1749 ± 35 | 1772 ± 46 | 1896 ± 37 | 2006 ± 49 | 1978 ± 38 | 0.0004 | 0.2 |
| Gender | n (%) female | 1,642 (50.0%) | 3,981 (49.4%) | 588 (55.2%) | 1,316 (63.6%) | 542 (52.9%) | 1,169 (50.7%) | 500 (40.6%) | 999 (44.1%) | 0.001 | <0.0001 |
| Education[e] | High school or less | NA | 3,355 (39.0%) | NA | 813 (34.7%) | NA | 858 (35.5%) | NA | 806 (35.3%) | NA | 0.3 |
| | Certificate or diploma | NA | 2,538 (34.5%) | NA | 687 (35.3%) | NA | 717 (31.9%) | NA | 670 (31.1%) | | |
| | University degree or higher | NA | 1,634 (25.9%) | NA | 530 (28.8%) | NA | 567 (31.7%) | NA | 650 (33.3%) | | |
| | Not stated | NA | 51 (0.6%) | NA | 14 (1.1%) | NA | 14 (0.8%) | NA | 15 (0.3%) | | |
| Marital status | Married or living common-law | NA | 4,313 (62.6%) | NA | 1,207 (66.8%) | NA | 1,341 (68.0%) | NA | 1,298 (65.4%) | NA | 0.3 |
| | Widowed, divorced or separated | NA | 1,612 (13.8%) | NA | 515 (16.2%) | NA | 457 (15.1%) | NA | 478 (15.3%) | | |
| | Single, never married | NA | 1,621 (23.3%) | NA | 313 (16.9%) | NA | 352 (16.8%) | NA | 353 (19.0%) | | |
| | Not stated | NA | 32 (0.4%) | NA | 9 (0.1%) | NA | 6 (0.1%) | NA | 12 (0.3%) | | |
| Current smoking status[f] | Daily or occasionally | 94 (2.1%) | 1840 (22.3%) | 12 (0.5%) | 292 (15.4%) | 21 (1.3%) | 309 (13.7%) | 17 (1.1%) | 259 (10.6%) | <0.0001 | 0.0002 |
| | Not at all | 1,545 (42.5%) | 5,727 (77.6%) | 258 (21.5%) | 1,746 (84.4%) | 377 (33.7%) | 1,841 (86.3%) | 439 (41.4%) | 1,876 (89.3%) | | |
| | Under 12 or no response | 1,666 (55.4%) | 11 (0.1%) | 807 (78.0%) | 6 (0.1%) | 687 (65.0%) | 6 (0.1%) | 645 (57.5%) | 6 (0.1%) | | |
| Overweight/ obesity[g] | n (%) yes | 824 (22.1%) | 3,330 (41.4%) | 183 (16.9%) | 847 (36.2%) | 204 (16.6%) | 945 (40.1%) | 229 (21.1%) | 893 (38.7%) | 0.06 | 0.4 |
| Income[h] | n (%) low income | 932 (27.2%) | 1,947 (23.3%) | 266 (25.9%) | 485 (20.7%) | 236 (22.2%) | 489 (22.4%) | 277 (26.5%) | 537 (20.8%) | 0.5 | 0.7 |
| Food insecurity | n (%) Moderately or severely food insecure | 443 (11.3%) | 854 (9.3%) | 137 (11.9%) | 146 (6.4%) | 124 (9.0%) | 156 (6.1%) | 110 (9.8%) | 153 (5.3%) | 0.6 | 0.05 |
| Immigrant to Canada | n (%) yes | 282 (9.1%) | 1,470 (27.0%) | 54 (6.4%) | 379 (23.3%) | 54 (6.8%) | 430 (29.0%) | 102 (11.5%) | 601 (33.3%) | 0.2 | 0.9 |
| Official language in which respondent can converse | English only | 2251 (64.2%) | 5,535 (67.2%) | 763 (65.7%) | 1,527 (70.9%) | 782 (70.7%) | 1,617 (71.9%) | 796 (71.7%) | 1,739 (77.1%) | 0.01 | 0.2 |
| | French only | 369 (16.3%) | 635 (10.8%) | 120 (17.7%) | 156 (9.0%) | 86 (10.6%) | 152 (9.9%) | 80 (10.4%) | 86 (4.9%) | | |
| | Both English and French | 632 (18.1%) | 1,328 (20.9%) | 167 (13.6%) | 332 (18.4%) | 197 (16.2%) | 368 (17.4%) | 210 (16.8%) | 265 (16.0%) | | |
| | Neither English nor French | 50 (1.4%) | 68 (1.1%) | 26 (2.8%) | 23 (1.1%) | 18 (2.3%) | 15 (0.7%) | 13 (1.0%) | 41 (1.8%) | | |
| | Not stated | 3 (0.0%) | 12 (0.1%) | 1 (0.3%) | 6 (0.6%) | 2 (0.1%) | 4 (0.2%) | 2 (0.1%) | 10 (0.2%) | | |

*(Continued)*

**Table 1.** (Continued)

| | | No Whole Grain Food Intake (No-WG)[a] | | Low Whole Grain Food Intake (Low-WG)[a] | | Mid-Whole Grain Food Intake (Mid-WG)[a] | | High-Whole Grain Food Intake (High-WG)[a] | | p value[b] | |
| --- | --- | --- | --- | --- | --- | --- | --- | --- | --- | --- | --- |
| | | Children n = 3,305 | Adults n = 7,578 | Children n = 1,077 | Adults n = 2,044 | Children n = 1,085 | Adults n = 2,156 | Children n = 1,101 | Adults n = 2,141 | Children | Adults |
| Supplement use | n (%) yes | 1090 (34.5%) | 3,235 (40.7%) | 428 (38.4%) | 1,090 (51.3%) | 457 (44.7%) | 1,139 (51.6%) | 433 (41.0%) | 1,140 (56.8%) | 0.004 | <0.0001 |
| Meeting physical activity guidelines[i] | n (%) yes | 934 (37.3%) | 3242 (43.5%) | 261 (43.7%) | 870 (49.1%) | 339 (51.5%) | 928 (43.9%) | 377 (46.5%) | 1006 (46.0%) | 0.001 | 0.3 |

Data are based on the Canadian Community Health Survey (CCHS) 2015 and are presented as mean ± standard error for age and as n and percent for categorical variables.

[a]CCHS 2015 respondents were stratified according to whole grain intake: those in the "no whole grain intake" (no-WG) group reported consuming no whole grain foods on a single 24-hour dietary recall. The remaining participants that consumed whole grains were divided according to age-specific tertiles into low- mid- and high- whole grain foods intake (low-WG, mid-WG, and high-WG).

[b]For age, statistical significance was determined by testing for a linear trend across whole grain intake groups. For categorical variables, statistical significance was determined using a chi$^2$ test. A $p < 0.05$ was considered statistically significant.

[c]Servings of whole grain foods was a pre-defined variable in the CCHS 2015 database.

[d] We calculated the ratio of the mean of Whole Grains (tiers 1–3 + tier 3) to total grain intake (defined as refined grains tier 1–3 + refined grains tier 4 + whole grains tier 1–3 + whole grains tier 4).

[e]We only included results on educational attainment for participants 19 years of age and older only.

[f]Smoking status were only reported for participants 12 years of age and older. Respondents in "not at all" category may include former smokers.

[g]Overweight and obesity were defined according to World Health Organization BMI cutoffs for adults and BMI z-score cut-offs for children.

[h]The variable low income was created according to the definition in the 2016 Canadian Census Dictionary and considers family gross income and family size.

[i]Meeting physical activity recommendations is reported only for children 6 years and older. For children, CCHS reported the percent of children 6 years of age and older that reported 60 minutes or more of physical activity per day. For adults, CCHS reported the percent of adults that reported 150 minutes or more per week of physical activity.

In addition to fiber, we also found higher consumption of several key nutrients including vitamin B$_6$ (in children only), thiamin (adults only), potassium, zinc (adults only), calcium, iron, and magnesium. We found no linear trend across the four groups of whole grain intake for either children or adults for vitamin B$_{12}$, vitamin C, vitamin D, niacin, vitamin A, or riboflavin. In contrast, folic acid intake was 21% lower for children and 36% for adults in the high-WG group compared to the no-WG group.

Saturated fat intake did not differ by whole grain intake among children. Total fat in grams was not significant different for children, but when expressed as a percent of energy decreased across increasing WG intake groups ($p < 0.0001$ for linear trend). Among adults, saturated fat (from 23.3 g/d in the no-WG group to 20.5 g/d in the high-WG group, $p < 0.0001$ for linear trend) and total fat (from 70.0 g/d in the no-WG group to 63.9 g/d in the high-WG group, $p < 0.0001$ for linear trend) intake decreased with increasing whole grain intake in adults. Total fat as a percent of energy also decreased across groups of increasing WG intake in adults. There was no significant trend in sodium intake across intakes of whole grain for both children and adults. Higher intake of numerous nutrients to encourage and no increase or lower intake of nutrients to limit (saturated fat, sodium, and sugar) resulted significant differences in diet quality across groups of WG intake (Fig 1) with pair-wise post-hoc tests indicating that all of the WG intake groups (low, mid, and high), for both children and adults, had significantly higher diet quality compared to the no-WG group. For adults, there were no significant differences in diet quality between the low-, mid- and high-WG intake groups; however, for children, the high-WG intake group had significantly higher diet quality compared to the

**Table 2. Adjusted daily nutrient intakes for Canadian children and adults stratified by whole grain food intake.**

| | No Whole Grain Food Intake (No-WG)[a] | | Low Whole Grain Food Intake (Low-WG)[a] | | Mid-Whole Grain Food Intake (Mid-WG)[a] | | High-Whole Grain Food Intake (High-WG)[a] | | p value for linear trend[b] | |
|---|---|---|---|---|---|---|---|---|---|---|
| | Children, n = 3,305 | Adults, n = 7,578 | Children, n = 1,077 | Adults, n = 2,044 | Children, n = 1,085 | Adults, n = 2,156 | Children, n = 1,101 | Adults, n = 2,141 | Children | Adults |
| Carbohydrates, g | 243 ± 3 | 221 ± 2 | 243 ± 3 | 222 ± 4 | 254 ± 4 | 225 ± 4 | 260 ± 4 | 244 ± 3 | <0.0001 | <0.0001 |
| Carbohydrate, % kcal | 53.2 ± 0.5 | 47.6 ± 0.6 | 53.2 ± 0.7 | 48.1 ± 1.0 | 54.5 ± 0.6 | 48.7 ± 0.7 | 55.6 ± 0.5 | 52.6 ± 0.8 | <0.0001 | <0.0001 |
| Fiber, g | 13.3 ± 0.3 | 14.6 ± 0.4 | 14.2 ± 0.3 | 17.5 ± 0.4 | 16.5 ± 0.4 | 18.7 ± 0.3 | 20.2 ± 0.4 | 23.7 ± 1.2 | <0.0001 | <0.0001 |
| Total sugar, g | 106.1 ± 3.3 | 90.3 ± 2.0 | 108.2 ± 3.3 | 91.9 ± 2.1 | 111.2 ± 3.2 | 87.7 ± 2.5 | 104.3 ± 3.5 | 87.6 ± 4.8 | 0.9 | 0.2 |
| Total fat, g | 65.2 ± 1.0 | 70.0 ± 1.2 | 65.1 ± 1.2 | 67.9 ± 1.0 | 65.2 ± 1.7 | 69.8 ± 1.2 | 62.7 ± 1.5 | 63.9 ± 1.8 | 0.2 | <0.0001 |
| Total fat, %kcal | 31.4 ± 0.4 | 32.4 ± 0.4 | 31.0 ± 0.5 | 31.9 ± 0.6 | 30.4 ± 0.6 | 32.0 ± 0.5 | 28.8 ± 0.5 | 29.4 ± 0.6 | <0.0001 | <0.0001 |
| Saturated fat, g | 23.3 ± 0.5 | 23.3 ± 0.8 | 23.4 ± 0.5 | 22.3 ± 0.5 | 22.7 ± 0.6 | 22.7 ± 0.4 | 22.0 ± 0.5 | 20.5 ± 0.6 | 0.08 | 0.0001 |
| Protein, g | 69.1 ± 1.5 | 77.8 ± 1.5 | 72.0 ± 2.3 | 78.7 ± 1.6 | 69.3 ± 2.0 | 78.5 ± 2.0 | 72.8 ± 2.0 | 77.6 ± 1.5 | 0.1 | 0.9 |
| Protein, %kcal | 15.3 ± 0.3 | 16.8 ± 0.2 | 15.7 ± 0.5 | 17.2 ± 0.3 | 15.0 ± 0.5 | 16.9 ± 0.3 | 15.5 ± 0.3 | 16.7 ± 0.3 | 0.9 | 1.0 |
| Vitamin $B_{12}$, µg | 3.81 ± 0.23 | 4.15 ± 0.19 | 3.91 ± 0.18 | 4.65 ± 0.53 | 3.58 ± 0.18 | 4.05 ± 0.18 | 3.48 ± 0.20 | 3.72 ± 0.34 | 0.1 | 0.3 |
| Vitamin $B_6$, µg | 1.39 ± 0.05 | 1.60 ± 0.04 | 1.44 ± 0.06 | 1.70 ± 0.04 | 1.43 ± 0.07 | 1.71 ± 0.05 | 1.64 ± 0.06 | 1.77 ± 0.10 | 0.0003 | 0.008 |
| Vitamin C, mg | 117 ± 5 | 101 ± 4 | 116 ± 7 | 103 ± 5 | 118 ± 11 | 110 ± 8 | 112 ± 6 | 109 ± 7 | 0.6 | 0.1 |
| Folate, µg | 454 ± 12 | 443 ± 12 | 460 ± 16 | 424 ± 10 | 415 ± 12 | 406 ± 10 | 410 ± 14 | 400 ± 11 | 0.0004 | <0.0001 |
| Folic acid, µg | 147 ± 6 | 123 ± 3 | 151 ± 9 | 109 ± 4 | 124 ± 7 | 94 ± 4 | 116 ± 7 | 79 ± 4 | <0.0001 | <0.0001 |
| Vitamin D, µg | 4.94 ± 0.26 | 5.02 ± 0.31 | 5.79 ± 0.31 | 5.46 ± 0.45 | 5.26 ± 0.25 | 5.47 ± 0.32 | 5.35 ± 0.32 | 5.32 ± 0.33 | 0.1 | 0.1 |
| Niacin, mg | 33.1 ± 0.8 | 38.4 ± 1.0 | 33.9 ± 1.2 | 38.5 ± 0.9 | 32.7 ± 1.1 | 38.4 ± 1.5 | 34.7 ± 1.0 | 37.7 ± 1.1 | 0.2 | 0.4 |
| Vitamin A, µg RAE | 582 ± 36 | 617 ± 37 | 633 ± 39 | 702 ± 42 | 582 ± 31 | 676 ± 28 | 570 ± 48 | 651 ± 30 | 0.8 | 0.4 |
| Riboflavin, mg | 1.82 ± 0.04 | 1.91 ± 0.04 | 1.87 ± 0.04 | 1.93 ± 0.06 | 1.77 ± 0.04 | 1.88 ± 0.04 | 1.77 ± 0.06 | 1.84 ± 0.04 | 0.3 | 0.05 |
| Thiamin, mg | 1.58 ± 0.03 | 1.50 ± 0.05 | 1.57 ± 0.04 | 1.52 ± 0.06 | 1.57 ± 0.06 | 1.54 ± 0.05 | 1.76 ± 0.07 | 1.72 ± 0.03 | 0.04 | 0.002 |
| Sodium, mg | 2568 ± 45 | 2783 ± 53 | 2573 ± 49 | 2689 ± 54 | 2544 ± 53 | 2705 ± 79 | 2578 ± 61 | 2748 ± 136 | 1.0 | 0.4 |
| Potassium, mg | 2293 ± 31 | 2645 ± 47 | 2363 ± 42 | 2780 ± 53 | 2418 ± 54 | 2780 ± 57 | 2467 ± 48 | 2852 ± 89 | 0.0002 | 0.0001 |
| Zinc, mg | 8.8 ± 0.2 | 10.1 ± 0.1 | 9.3 ± 0.3 | 10.7 ± 0.5 | 9.1 ± 0.4 | 10.5 ± 0.3 | 9.8 ± 0.3 | 10.9 ± 0.2 | 0.004 | 0.0001 |
| Calcium, mg | 873 ± 28 | 755 ± 26 | 979 ± 25 | 826 ± 27 | 945 ± 28 | 806 ± 23 | 971 ± 38 | 807 ± 31 | 0.001 | 0.001 |
| Iron, mg | 11.6 ± 0.2 | 11.7 ± 0.2 | 12.5 ± 0.4 | 12.6 ± 0.3 | 12.3 ± 0.4 | 12.5 ± 0.4 | 13.6 ± 0.4 | 13.2 ± 0.2 | <0.0001 | <0.0001 |
| Magnesium, mg | 234 ± 5 | 277 ± 4 | 255 ± 8 | 307 ± 6 | 268 ± 6 | 325 ± 6 | 311 ± 7 | 367 ± 6 | <0.0001 | <0.0001 |

RAE, retinol activity equivalents. [a]Data are based on the Canadian Community Health Survey (CCHS) 2015 and are presented as least squares mean ± standard error. Results (except for energy intake which was unadjusted) were adjusted for energy, gender, age, BMI category, income, and supplement use.

[a]CCHS 2015 respondents were stratified according to whole grain intake: those in the "no whole grain intake" (No-WG) group reported consuming no whole grain foods on a single 24-hour dietary recall. The remaining participants that consumed whole grains were divided according to age-specific tertiles into low- middle- and high-whole grain foods intake (low-WG, mid-WG, high-WG).

[b]Statistical significance was determined based on linear trends across whole grain intake groups for children and adults (separately) and a p≤0.002, after applying a Bonferroni correction for multiple comparisons (0.05/22 nutrients = 0.00227), was considered statistically significant.

low- and mid-WG intake groups. There was no significant difference between the low-WG and mid-WG intake groups for children. Unadjusted NRF 9.3 values can be found in S5 Table and the p values for the post-hoc pairwise comparisons can be found in S6 Table.

## Food group intake

Intake of all food groups in tiers 1–3 were highest in the high-WG group and lowest in the no-WG group for both children and adults (Table 3). This was due to higher intake (based on linear trends across all 4 whole grain intake groups) of fruits tiers 1–3 (adults); legumes, nuts and seeds tiers 1–3 (adults); and whole grains tiers 1–3 (children and adults) for the high-WG

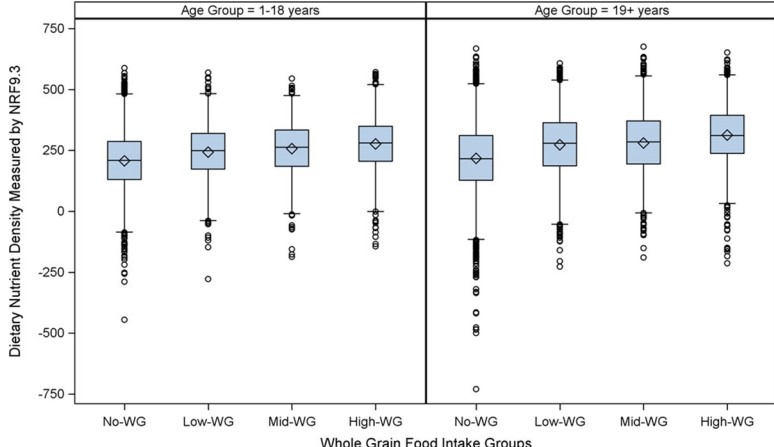

**Fig 1. Diet quality, measured by the nutrient rich food index 9.3, among Canadian children and adults stratified by whole grain food intake.** Data are from the Canadian Community Health Survey (CCHS) 2015. The left panel shows results for children 1 to 18 years and the right panel shows results for adults 19 years and older. Data are presented as boxplots for the Nutrient Rich Food Index (NRF) 9.3 (displayed on the y-axis in arbitrary units), a measure of nutrient density adapted to the total diet that calculates the sum of the contribution to daily recommendations for nutrients to encourage (fiber, protein, vitamin D, vitamin C, iron, calcium, potassium, vitamin A, and magnesium) minus the sum of the contribution to daily recommendations for nutrients to limit (total sugar, sodium, and saturated fat) per 2000 kcals. Participants were stratified according to whole grain (WG) food consumption reported on a single 24 hr dietary recall. The no-WG group reported consuming no whole grain foods (children n = 3,305; adults n = 7,578); The remaining participants that consumed whole grains were divided according to age-specific tertiles into low-whole grain (low-WG; children n = 3,305; adults n = 7,578), mid-whole grain (mid-WG, children n = 1,077; adults n = 2,044) and high-whole grain (high-WG; children n = 1,101; adults 2,141) food intake. The boxplot displays the median (center line of the box), the mean (diamond), the 25th percentile (lower line of the box), the 75th percentile (upper line of the box), the 5st percentile (lower end of the whisker), the 95th percentile (upper end of the whisker), and outliers (open circles). Statistical significance was tested using linear regression adjusting for gender, age, BMI, low-income status, supplement use, and total dietary energy intake and we found a statistically significant ($p < 0.0001$ for both children and adults) relationship between WG consumption status and diet quality, as measured by the NRF 9.3, as well as a statistically significant trend across groups of WG intake ($p < 0.0001$).

group compared to the no- and low-WG groups. The high-WG group also had lower intake of refined grains tiers 1–3 (children and adults), and meat and poultry tiers 1–3 (adults) and no difference across all 4 groups for the other tier 1–3 food groups.

Intake of tier 4 foods were generally much less (on a per gram basis) than tiers 1–3 foods and total tier 4 food intake was significantly lower for the high-WG group compared to the no-WG group for adults only. Intake of refined grains tier 4 (children), meat and poultry tier 4 (adults), and processed meat tier 4 (adults) were significantly lower with increasing whole grain intake. Foods that were considered discretionary according to the CFG were consumed, on a per gram basis, 48% less among adults in the high-WG group compared to the no-WG group ($p < 0.0001$ linear trend across 4 whole grain intake groups). There was no significant difference in discretionary food intake among children according to whole grain intake group. Unadjusted mean food group intake can be found in S7 Table.

## Food sources of whole grain

Top sources of whole grain foods were similar across the three whole grain intake groups for children and adults (i.e. the low-, Mid-, and High-WG groups, Tables 4 and 5). The top two sources of whole grain across all groups were 1) whole grain oat and high fiber breakfast cereal and 2) whole grain and whole wheat bread, except for the low-WG group for children whose top source of whole grain was other breakfast cereal. Despite the same top two sources across

**Table 3. Adjusted Canadian food guide food group intake for children and adults stratified by whole grain food intake.**

| Food Groups by Canada Food Guide Tiers[c] | No Whole Grain Food Intake (No-WG)[a] | | Low Whole Grain Food Intake (Low-WG)[a] | | Mid-Whole Grain Food Intake (Mid-WG)[a] | | High-Whole Grain Food Intake (High-WG)[a] | | p value for linear trend[b] | |
|---|---|---|---|---|---|---|---|---|---|---|
| | Children, n = 3,305 | Adults, n = 7,578 | Children, n = 1,077 | Adults, n = 2,044 | Children, n = 1,085 | Adults, n = 2,156 | Children, n = 1,101 | Adults, n = 2,141 | Children | Adults |
| Whole Grains Tier 1–3, g/d | 0 ± 0 | 0 ± 0 | 17 ± 2 | 24 ± 3 | 51 ± 1 | 66 ± 3 | 154 ± 6 | 213 ± 14 | <0.0001 | <0.0001 |
| Whole Grains Tier 4, g/d | 0 ± 0 | 0 ± 0 | 3 ± 1 | 2 ± 0 | 4 ± 1 | 3 ± 1 | 8 ± 1 | 3 ± 1 | <0.0001 | <0.0001 |
| Fruit tiers 1–3, g/d | 271 ± 12 | 205 ± 8 | 273 ± 18 | 224 ± 11 | 302 ± 26 | 240 ± 12 | 294 ± 18 | 240 ± 11 | 0.07 | 0.0001 |
| Fruit tier 4, g/d | 0 ± 0 | 0 ± 0 | 0 ± 0 | 0 ± 0 | 0 ± 0 | 0 ± 0 | 0 ± 0 | 0 ± 0 | 1.0 | 1.0 |
| Vegetables tiers 1–3, g/d | 149 ± 6 | 219 ± 7 | 148 ± 12 | 219 ± 10 | 141 ± 10 | 222 ± 9 | 145 ± 7 | 227 ± 20 | 0.4 | 0.7 |
| Vegetables tier 4, g/d | 3 ± 0 | 5 ± 1 | 2 ± 1 | 4 ± 1 | 2 ± 1 | 5 ± 2 | 4 ± 2 | 2 ± 1 | 0.7 | 0.2 |
| Refined grains tiers 1–3, g/d | 206 ± 7 | 187 ± 9 | 185 ± 9 | 155 ± 8 | 159 ± 8 | 129 ± 14 | 140 ± 8 | 116 ± 6 | <0.0001 | <0.0001 |
| Refined grains tier 4, g/d | 45 ± 7 | 24 ± 9 | 41 ± 9 | 21 ± 8 | 35 ± 8 | 23 ± 14 | 27 ± 8 | 18 ± 6 | 0.001 | 0.04 |
| Dairy products and alternatives tiers 1–3, g/d | 326 ± 18 | 214 ± 17 | 390 ± 18 | 261 ± 12 | 377 ± 17 | 250 ± 13 | 377 ± 17 | 242 ± 15 | 0.07 | 0.004 |
| Dairy products and alternatives tier 4, g/d | 5 ± 1 | 7 ± 1 | 5 ± 1 | 5 ± 1 | 6 ± 2 | 5 ± 1 | 3 ± 1 | 5 ± 2 | 0.2 | 0.009 |
| Meat and poultry tiers 1–3, g/d | 62 ± 4 | 84 ± 5 | 64 ± 7 | 84 ± 5 | 52 ± 6 | 77 ± 5 | 56 ± 6 | 66 ± 5 | 0.08 | 0.0005 |
| Meat and poultry tier 4, g/d | 9 ± 2 | 12 ± 2 | 7 ± 2 | 7 ± 2 | 8 ± 3 | 5 ± 4 | 6 ± 3 | 6 ± 2 | 0.1 | 0.0006 |
| Processed meats tiers 1–3, g/d | 12 ± 2 | 11 ± 1 | 11 ± 2 | 11 ± 2 | 12 ± 2 | 12 ± 1 | 12 ± 3 | 13 ± 3 | 1.0 | 0.4 |
| Processed meats tier 4, g/d | 10 ± 1 | 9 ± 1 | 12 ± 2 | 6 ± 2 | 10 ± 2 | 7 ± 3 | 6 ± 1 | 4 ± 1 | 0.07 | 0.0003 |
| Fish and shellfish tiers 1–3, g/d | 7 ± 1 | 18 ± 5 | 7 ± 2 | 22 ± 7 | 6 ± 1 | 20 ± 5 | 9 ± 2 | 19 ± 5 | 0.9 | 0.4 |
| Fish and shellfish tier 4, g/d | 1 ± 0 | 4 ± 1 | 1 ± 1 | 3 ± 1 | 0 ± 0 | 4 ± 1 | 2 ± 1 | 2 ± 2 | 0.5 | 0.09 |
| Legumes, nuts and seeds tiers 1–3, g/d | 13 ± 2 | 23 ± 3 | 11 ± 2 | 31 ± 5 | 19 ± 3 | 28 ± 3 | 17 ± 3 | 37 ± 4 | 0.08 | <0.0001 |
| Legumes, nuts and seeds tier 4, g/d | 1 ± 0 | 1 ± 0 | 0 ± 0 | 3 ± 1 | 1 ± 1 | 0 ± 1 | 1 ± 0 | 2 ± 1 | 0.9 | 0.4 |
| Eggs tiers 1–3, g/d | 14 ± 1 | 21 ± 2 | 15 ± 2 | 21 ± 2 | 15 ± 2 | 25 ± 2 | 13 ± 2 | 22 ± 2 | 0.9 | 0.4 |
| Eggs tier 4, g/d | 1 ± 0 | 0 ± 0 | 0 ± 0 | 0 ± 1 | 0 ± 0 | 0 ± 0 | 0 ± 0 | 0 ± 0 | 0.8 | 0.3 |
| No CFG—Discretionary foods, g/d | 191 ± 18 | 315 ± 14 | 164 ± 19 | 251 ± 16 | 175 ± 18 | 228 ± 17 | 152 ± 14 | 163 ± 17 | 0.07 | <0.0001 |
| No CFG—Other foods and recipes, g/d | 767 ± 37 | 1316 ± 54 | 809 ± 46 | 1410 ± 45 | 803 ± 41 | 1434 ± 42 | 865 ± 57 | 1435 ± 46 | 0.03 | 0.09 |
| No CFG—Foods not classified, g/d | 562 ± 16 | 713 ± 23 | 582 ± 23 | 729 ± 30 | 587 ± 21 | 729 ± 41 | 551 ± 20 | 723 ± 26 | 0.9 | 0.6 |
| **Total Tier 1–3 foods, g/d** | 1061 ± 20 | 979 ± 17 | 1121 ± 22 | 1052 ± 20 | 1132 ± 37 | 1068 ± 22 | 1192 ± 28 | 1195 ± 20 | <0.0001 | <0.0001 |
| **Total Tier 4 foods, g/d** | 75 ± 7 | 62 ± 3 | 71 ± 6 | 52 ± 9 | 66 ± 6 | 52 ± 5 | 57 ± 6 | 42 ± 4 | 0.01 | <0.0001 |
| **Total non-CFG foods, g/d** | 1520 ± 31 | 2345 ± 61 | 1554 ± 39 | 2391 ± 46 | 1566 ± 41 | 2391 ± 53 | 1568 ± 59 | 2321 ± 67 | 0.3 | 1.0 |

CFG, Canada Food Guide; WG, whole grains. Data are based on the Canadian Community Health Survey (CCHS) 2015 and are presented as least squares mean ± standard error. The units for all results are in grams. Results were adjusted for energy, gender, age, BMI category, income, and supplement use.

[a]CCHS 2015 respondents were stratified according to whole grain intake: those in the "no whole grain intake" (No-WG) group reported consuming no whole grain foods on a single 24-hour dietary recall. The remaining participants that consumed whole grains were divided according to age-specific tertiles into low- middle- and high-whole grain foods intake (low-WG, mid-WG, high-WG).

[b]Statistical significance was determined based on linear trends across whole grain intake groups for children and adults (separately) and a p≤0.002, after applying a Bonferroni correction for multiple comparisons (0.05/24 food groups = 0.00208), was considered statistically significant.

[c]The four tiers indicate whether a food item is aligned (tiers 1 & 2), partially aligned (tier 3) or not aligned (tier 4) with the 2007 Canadian Food Guide.

groups, there were large differences in the amounts of these foods consumed: the high-WG group consumed an average of 95.1 g/d for children and 135.3 g/d for adults of these two foods while the mid-WG group consumed 38.2 and 55.4 g/d for children and adults respectively and the low-WG group children consumed 12.8 g/d (of other breakfast cereal and whole grain, oats and high fibre breakfast cereals) while adults consumed 18.9 g/d. The remaining top 10 list included rice, other

**Table 4. Top ten food sources of whole grain for Canadian children stratified by whole grain intake.**

| Ranking | Low WG Intake Group (Low-WG)[a] | | Mid WG Intake Group (Mid-WG)[a] | | High WG Intake Group (High-WG)[a] | |
|---|---|---|---|---|---|---|
| | BNS Food Group[b] | Mean Intake g/d ± SE | BNS Food Group[b] | Mean Intake g/d ± SE | BNS Food Group[b] | Mean Intake g/d ± SE |
| 1 | Other breakfast cereal | 6.4 ± 0.6 | Whole grain and whole wheat bread | 27.4 ± 1.8 | Whole grain, oat, and high fibre breakfast cereal | 56.4 ± 5.9 |
| 2 | Whole grain, oats, and high fibre breakfast cereals | 6.4 ± 0.9 | Whole grain, oat, and high fibre breakfast cereal | 10.8 ± 1.1 | Whole grain and whole wheat bread | 38.7 ± 2.6 |
| 3 | Whole grain and whole wheat bread | 2.6 ± 0.4 | Rolls, bagels, pita bread, croutons, dumplings, matzo, tortilla | 3.4 ± 0.7 | Rice | 18.5 ± 3.1 |
| 4 | Salty and high-fat snacks (incl Tortilla chips) | 1.2 ± 0.1 | Other breakfast cereal | 3.3 ± 0.8 | Rolls, bagels, pita bread, croutons, dumplings, matzo, tortilla | 16.3 ± 3.3 |
| 5 | Sweet snacks, sugar, candies | 0.7 ± 0.2 | Rice | 2.4 ± 1.1 | Other breakfast cereal | 6.6 ± 1.5 |
| 6 | Rolls, bagels, pita bread, croutons, dumplings, matzo, tortilla | 0.6 ± 0.2 | Salty and high-fat snacks (including tortilla chips) | 1.8 ± 0.4 | Salty and high-fat snacks (including tortilla chips) | 6.0 ± 1.0 |
| 7 | Plain popcorn and pretzels | 0.4 ± 0.1 | Pancakes and waffles | 1.4 ± 0.4 | Pasta | 5.1 ± 2.1 |
| 8 | Cereal grains and flours | 0.3 ± 0.1 | Sweet snacks, sugar, candies | 1.3 ± 0.4 | Pancakes and waffles | 3.9 ± 1.5 |
| 9 | Crackers and crispbreads | 0.2 ± 0.1 | Cereal grains and flours | 0.6 ± 0.3 | Cereal grains and flours | 3.3 ± 0.8 |
| 10 | Rice | 0.2 ± 0.1 | Granola bar | 0.5 ± 0.4 | Sweet snacks, sugar, candies | 2.1 ± 0.6 |
| Total intake from top 10 sources of WG foods (% of total WG intake) | | 19.1 ± 0.8 (99.0%) | | 52.9 ± 1.6 (97.1%) | | 156.9 ± 5.3 (96.2%) |

BNS, Bureau of Nutritional Sciences; WG, whole grains. Data are based on the Canadian Community Health Survey (CCHS) 2015 and are presented as least squares mean ± standard error. Results were adjusted for energy, gender, age, BMI category, income, and supplement use.

[a]CCHS 2015 respondents were stratified according to whole grain intake: those in the "no whole grain intake" (No-WG) group reported consuming no whole grain foods on a single 24-hour dietary recall. The remaining participants that consumed whole grains were divided according to age-specific tertiles into low- middle- and high-whole grain foods intake (low-WG, mid-WG, high-WG).

[b]The BNS food groups were developed by the Bureau of Nutritional Sciences (BNS) at Health Canada based on the British and American food group systems. We used the BNS food groups to report the top sources of whole grain in order to provide more granularity on the types of foods that were consumed within the CFG whole grain tier 1–4 food group.

bread products such as rolls and bagels, other breakfast cereals, salty snacks such as tortilla chips, cereal grains and flours, pasta, and sweet snacks (Tables 4 and 5). Whole grain foods were broadly similar across whole grain intake groups for children, with only the low-WG having plain popcorn and pretzels and crackers and crispbreads in the top 10. For adults, rice and pasta were not among the top 10 whole grain foods in the low-WG group while foods such as crackers and crispbreads, granola bars, and muffins and English muffins were. Based on these top 10 foods, which account for over 95% of total whole grain food intake, the high-WG groups had an over 680% higher intake of these foods compared to the low-WG group.

## Discussion

We found that higher whole grain intake in children and adults was associated with higher intake of several key nutrients, food groups, and improved diet quality. These findings are supported by previous results in both Canada [31] and other regions US [32], and UK [33]. Our study extended these previous results by providing detailed information on nutrient and food group intakes in Canada but also reporting the top food sources of whole grain in the Canadian diet.

Fifty percent of children and 54% of adults in this Canadian dataset consumed no whole grain foods on their single 24-hour dietary recall. Among the children and adults that reported

**Table 5. Top ten food sources of whole grain for Canadian adults stratified by whole grain intake.**

| Ranking | Low WG Intake Group (Low-WG)[a] | | Mid WG Intake Group (Mid-WG)[a] | | High WG Intake Group (High-WG)[a] | |
|---|---|---|---|---|---|---|
| | BNS Food Group[b] | Mean Intake g/d ± SE | BNS Food Group[b] | Mean Intake g/d ± SE | BNS Food Group[b] | Mean Intake g/d ± SE |
| 1 | Whole grain, oat, and high fibre breakfast cereal | 11.0 ± 0.6 | Whole grain and whole wheat bread | 40.3 ± 1.6 | Whole grain, oat, and high fibre breakfast cereal | 89.8 ± 12.4 |
| 2 | Whole grain and whole wheat bread | 7.9 ± 0.6 | Whole grain, oat, and high fibre breakfast cereal | 15.1 ± 1.8 | Whole grain and whole wheat bread | 45.5 ± 2.8 |
| 3 | Other breakfast cereal | 2.7 ± 0.4 | Rolls, bagels, pita bread, croutons, dumplings, matzo, tortilla | 4.9 ± 1.1 | Rice | 35.8 ± 9.2 |
| 4 | Salty and high-fat snacks (including tortilla chips) | 1.3 ± 0.1 | Rice | 2.9 ± 0.7 | Rolls, bagels, pita bread, croutons, dumplings, matzo, tortilla | 26.1 ± 14.5 |
| 5 | Crackers and crispbreads | 1.1 ± 0.3 | Salty and high fat snacks (incl Tortilla chips) | 1.9 ± 0.4 | Cereal grains and flours | 9.3 ± 2.0 |
| 6 | Rolls, bagels, pita bread, croutons, dumplings, matzo, tortilla | 1.0 ± 0.4 | Other breakfast cereal | 1.2 ± 0.3 | Pasta | 4.3 ± 1.2 |
| 7 | Sweet snacks, sugar, candies | 1.0 ± 0.2 | Cereal grains and flours | 1.1 ± 0.4 | Salty and high-fat snacks (including tortilla chips) | 2.1 ± 0.5 |
| 8 | Granola bars | 0.5 ± 0.2 | Pasta | 0.7 ± 0.3 | Plain popcorn and pretzels | 1.5 ± 0.9 |
| 9 | Muffins and English muffins | 0.5 ± 0.4 | Other whole grain breads | 0.7 ± 0.2 | Sweet snacks, sugar, candies | 0.9 ± 0.4 |
| 10 | Cereal grains and flours | 0.4 ± 0.2 | Sweet snacks, sugar, candies | 0.7 ± 0.2 | Muffins | 0.9 ± 0.4 |
| Total intake from top 10 sources of WG foods (% of total WG intake) | | 27.5 ± 0.6 (95.2%) | | 69.4 ± 0.7 (96.9%) | | 216.2 ± 9.7 (98.3%) |

BNS, Bureau of Nutritional Sciences; WG, whole grains. Data are based on the Canadian Community Health Survey (CCHS) 2015 and are presented as least squares mean ± standard error. Results were adjusted for energy, gender, age, BMI category, income, and supplement use.

[a]CCHS 2015 respondents were stratified according to whole grain intake: those in the "no whole grain intake" (No-WG) group reported consuming no whole grain foods on a single 24-hour dietary recall. The remaining participants that consumed whole grains were divided according to age-specific tertiles into low- middle- and high-whole grain foods intake (low-WG, mid-WG, high-WG).

[b]The BNS food groups were developed by the Bureau of Nutritional Sciences (BNS) at Health Canada based on the British and American food group systems. We used the BNS food groups to report the top sources of whole grain in order to provide more granularity on the types of foods that were consumed within the CFG whole grain tier 1–4 food group.

consuming some whole grain foods, there was a large range in intake with those in the low-WG intake group consuming less than one serving/day on average and only 16% (for children) or 26% (for adults) of their total grain intake coming from whole grains while those in the high-WG group consumed over 3 servings/day of whole grain food and over 50% of their grain intake from whole grains.

Fiber intake was markedly increased across groups with increasing WG intake. Fiber has been identified in the U.S. as a nutrient of public health concern due to its widespread under-consumption and implications in human health [34]. The consumption of more WG foods appears to be an important strategy to increase fiber for both Canadian children and adults. Other food groups may also be contributing to fiber intake as we saw a positive trend for higher fruit and legume, nut and seed intake in adults (but not children) across the four groups of WG intake. However, legume, nut and seed intake was overall much lower than whole grain intake: adults in the high-WG group reported consuming 39g/day of legume, nuts and seeds compared to 216g/d of whole grains. Also, the difference in fruit intake between the no-WG group and high-WG group was 35g compared to a difference in whole grain intake between the two groups of 216g/d. Therefore, while multiple food groups, like fruits, vegetables,

legumes, nuts and seeds contribute to fiber intake, in this study, we saw large differences in fiber intake by WG consumption status.

In addition to fiber, we also found higher consumption of several key nutrients. Despite the generally higher nutrient intake in the high-WG group, folate and folic acid intakes were highest in the no-WG group for both children and adults, likely due to the mandatory enrichment of refined grains with this nutrient. The 2015 and 2020 Dietary Guidelines for Americans recommend that individuals who choose to consume all of their grains as whole grains should consume fortified whole grains, such as fortified ready-to-eat cereal, to ensure adequate folic acid intake [34,35]. Similar advice is provided by Health Canada [36]. This is particularly important for women who are or may become pregnant for the prevention of neural tube defects in their infants [36]. Similar pragmatic recommendations within the Canadian Dietary Guidelines may be useful to the Canada population. Associations between WG intake and increased intake of fiber, vitamins, and minerals may be partly explained by the fact that many whole grains foods are good sources of these nutrients, either inherently or through fortification.

A recent paper by Hosseini et al [31] used different methodology (propensity score matching) to report the association between categories of whole grain intake and nutrient intake and the prevalence of obesity. Patterns of nutrient intakes were broadly similar, with lower folate intake in the high WG consumption groups but higher intakes of magnesium, potassium, and fiber. However, they did not report the average intake of these nutrients across their WG intake groups (only the "average treatment effects on the treated" [ATE]). Our results are complementary to those of Hosseini by confirming their results related to the association between whole grain and nutrient intakes using a different methodological approach and extending them to include food groups and top sources of whole grain foods [31].

These findings highlight the importance of considering the overall nutrient density–that is the balance of nutrients and food groups to encourage and nutrients to limit–rather than only considering nutrients in isolation [37]. Some of the observed differences in nutrient intakes and diet quality may be due to the consumption of whole grain *per se* but are also likely due to the overall increased intake in other health-promoting food groups including fruits, vegetables, dairy, and nuts and seeds, particularly among foods classified by CCHS as tiers 1–3 (i.e. at least partially recommended by the 2007 CFG). Healthy dietary patterns should be built from nutrient dense foods that include key food groups such as whole grain, dairy, fruit, vegetables, and nuts and seeds [1,34].

Hosseini and colleagues also reported an association between dietary patterns with higher whole grain content and higher diet quality scores [31]. Several nutrient rich foods that are recommended by Canada's Food Guide, including whole grains as well as dairy, fruit, nuts and seeds, and beans and legumes, contain not only important nutrients, but can also contain saturated fat, sodium and/or added sugars which are either naturally occurring or added for palatability or functionality. Encouraging consumption of whole grain foods, even when they contain some sugar, sodium or saturated fat, may result in a higher quality diet for Canadians. Advice to choose nutrient dense foods which balance nutrients to encourage, food groups to encourage and nutrients to limit may be a more useful strategy for Canadians to build a healthy dietary pattern rather than focusing on calories, saturated fat, sodium and sugar alone [37,38].

We found that intake of recommended food groups tracked with whole grain intake. To our knowledge, this is the first publication to report the association between whole grain intake and association with other food groups. Tugault-Lafleur et al. [39] reported the change in food group intake from the 2004 Canadian dietary survey to the 2015 Canadian dietary survey and found that whole grain intake had not changed during this time period. However, they did not provide any further context on the association between whole grain intake and dietary intake

in the 2015 survey. These findings, combined with those of the current paper, highlight the important need to develop strategies to encourage Canadians to increase their whole grain consumption.

Novel to this study, we reported the top food sources of whole grain foods in Canadians' diet and 1) whole grain oat and high fiber breakfast cereal and 2) whole grain and whole wheat bread were the top two sources of whole grain for both children and adults. A potentially successful strategy to increase whole grain intake in Canada could be to encourage consumption of these foods, including whole grain breakfast cereals, bread, rice and pasta, in place of refined grains or other foods to maintain appropriate caloric intake. Other studies from the US [40], France [15] and Australia [41] have also reported ready-to-eat cereals and bread as among the top sources of whole grain in the diet.

## Strengths and limitations

This study used a large, nationally representative sample of the Canadian population that followed rigorous methodology. Detailed dietary intake was collected via a 24 hr dietary recall using a validated methodology. We examined ordinal groups of WG intake ranging from no WG intake to high WG intake and assessed statistical significance according to trends in our outcomes across these 4 groups, providing evidence that there may be a dose-response between WG intake and improved nutrient and food group intakes.

However, this study did have several limitations. First, due to the cross-sectional and observational study design, causation cannot be assigned. Despite using age-specific tertile cut-offs to define whole grain intake groups, we saw a significant association between whole grain intake and energy in children, but not adults, suggesting that potentially among children consuming more foods increased the likelihood of consuming more whole grain foods or represented underreporting. While misreporting of energy intake has been well-documented in self-reported dietary intakes, we found that very few participants in our study reported extreme calorie intakes below 200 kcal or above 8000 kcal. Tugault-Lafleur et al, using a different approach to classify mis-reporters, estimated in the CCHS 2015 that under-report of energy intake was 30% and overreporting of energy intake was 9% [39]. While there is no one "correct" way to identify mis-reporting in self-reported dietary intake studies, the potential for mis-reporting should be considered when interpreting these results [21]; however, this study is not attempting to estimate usual energy intake nor assess the association between dietary intake and health outcomes, where usual energy intake and energy misreporting would be more relevant.

Other potential confounding variables to consider that differed between our four WG intake groups include smoking status, supplement use and food insecurity. Several other lifestyle and socioeconomic variables status did not differ, however including education, overweight/obesity, and income. While we adjusted our outcomes for some of these important demographic characteristics, there remains the possibility of residual confounding, inherent to all observational research.

Second, this study included only dietary outcomes and did not extend to the potential health implications of nutrient and food group intake. We relied on a single 24 hr dietary recall which can provide reliable estimates of mean population intake of nutrients and food groups, but cannot estimate usual intake, which would be needed for studies examining health outcomes [42]. Our measure of diet quality did not include food groups, due to the lack of data in CCHS on quantitative intake of food groups from mixed dishes; however, we did rely upon a well-validated index of nutrient density to approximate dietary quality and complemented this information with average intake of CFG food groups.

Third, we examined intake of WG foods, instead of the total amount of WG specifically in the diet. Different WG foods contain different amount of WG; therefore, we may not be accurately capturing total WG intake as grams of WG food intake can be influenced by the non-WG ingredients and particularly impacted by the moisture content of the food. The CCHS did not disaggregate whole grain content from mixed dishes and we were unable to consider whole grain content from those dishes; however, there were few food codes from mixed dishes that included whole grains (or a similar term) in their description and only a small number of participants consumed these mixed dishes suggesting that our definition captured nearly all whole grain intake. We included WG foods that were both completely to partially aligned with the Canada Food Guide (tiers 1–3) and those that were not aligned (tier 4); however, intake of tier 4 WG foods was much lower than tiers 1–3 WG foods for all of our WG intake groups and this more inclusive definition of WG food would bias our results, if it had any impact, toward demonstrating less of a nutritional benefit to WG consumption. Future nutrient composition database should aim to quantify the amount of WG per food. This would also allow the whole grain intake of the Canadian population to be compared to whole grain intake recommendations, should these be developed in Canada (the currently Canadian Dietary Guidelines do not include quantitative whole grain or other food group recommendations) to determine the percent of the Canadian population meeting whole grain recommendations. While quantifying the whole grain content of each food and identifying an appropriate whole grain recommendation value for the Canadian population was outside the scope of this work, it is an important area of future research that merits further exploration.

## Conclusions

Despite whole grain intake being recommended in the most recent Canada Food Guide, there is little research on whole grain intake among Canadian children and adults or how whole grain intake is associated with overall dietary intake. We found that higher whole grain intake was associated with higher intake of key nutrients, particularly fiber, potassium and iron, and associated with higher diet quality. Intake of whole grain was associated with higher intake of other recommended food groups including fruits, dairy, and legumes, nuts and seeds. This research supports recommendations in Canadian Dietary Guidelines encouraging whole grain intake and shows that the top food sources of whole grain in the Canadian diet are high fiber breakfast cereals and breads. A potential strategy to encourage whole grain intake among Canadians could be to encourage intake of these two foods in place of refined grains or other foods. High whole grain intake is achieved by some Canadians children and adults and offering pragmatic solutions to encourage their consumption, considering the price, availability, and taste of food, may be important to ensuring that all Canadians consume more whole grains.

## Supporting information

**S1 Table. Individual food codes that included the term "whole wheat," "whole grain," or "wholemeal" that were not assigned to a Canada Food Guide (CFG) food group.** Data are from the Canadian Community Health Survey (CCHS) 2015.
(PDF)

**S2 Table. Food groups used in the current study based on Canada food guide food groups and tiers.**
(PDF)

**S3 Table. Daily values for nutrients included in the calculation of the nutrient rich food index 9.3.**
(PDF)

**S4 Table. Unadjusted daily nutrient intakes for Canadian children and adults stratified by whole grain food intake.** Data are based on the Canadian Community Health Survey (CCHS) 2015 and are presented as mean ± standard error. Results are unadjusted.
(PDF)

**S5 Table. Unadjusted mean nutrient rich food index 9.3 for Canadian children and adults stratified by whole grain food intake.** Data are based on the Canadian Community Health Survey (CCHS) 2015 and are presented as mean ± standard error in arbitrary units. Results are unadjusted.
(PDF)

**S6 Table. Pairwise differences in the nutrient-rich food index 9.3 for Canadian children and adults stratified by whole grain food intake.** Data are based on the Canadian Community Health Survey (CCHS) 2015 and are presented as the pairwise unadjusted difference in nutrient rich food index 9.3 scores (arbitrary units) and p-values based on the Tukey-Kramer post-hoc test from a multiple linear regression model that included total energy intake (categorical), age (continuous), gender (categorical), overweight/obesity status (yes/no), low-income (categorical), and supplement use (categorical) as covariates.
(PDF)

**S7 Table. Unadjusted Canadian food guide food group intake for children and adults stratified by whole grain food intake.** Data are based on the Canadian Community Health Survey (CCHS) 2015 and are presented as mean ± standard error. The results are unadjusted. The units for all results are in grams.
(PDF)

## Acknowledgments

We would like to acknowledge Megan Nechanicky and Jean-Michel Michno for their review and helpful edits to this manuscript based on their Canadian nutrition policy and statistical and data analytic expertise, respectively.

## Author Contributions

**Conceptualization:** Jessica Smith, Yong Zhu, Norton Holschuh.

**Formal analysis:** Yong Zhu, Neha Jain, Norton Holschuh.

**Methodology:** Jessica Smith, Yong Zhu, Neha Jain, Norton Holschuh.

**Project administration:** Jessica Smith.

**Supervision:** Jessica Smith.

**Visualization:** Jessica Smith, Neha Jain.

**Writing – original draft:** Jessica Smith.

**Writing – review & editing:** Jessica Smith, Yong Zhu, Neha Jain, Norton Holschuh.

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
