## [Decision Letter · Decision Letter 0]

13 Jan 2021

PONE-D-20-36067

Association between whole grain food intake in Canada and nutrient intake, food group intake and diet quality: Findings from the 2015 Canadian Community Health Survey

PLOS ONE

Dear Dr. Smith,

Thank you for submitting your manuscript to PLOS ONE. After careful consideration, we feel that it has merit but does not fully meet PLOS ONE’s publication criteria as it currently stands. Therefore, we invite you to submit a revised version of the manuscript that addresses the points raised during the review process.

We look forward to receiving your revised manuscript.

Kind regards,

Jimmy Louie

Academic Editor

PLOS ONE

Journal Requirements:

2.Thank you for stating the following in the Competing Interests section:

"All authors have read the journal's policy and the authors of have the following competing interests: All authors are employed by General Mills, a food manufacturer."

3. In statistical methods, please refer to any post-hoc corrections to correct for multiple comparisons during your statistical analyses. If these were not performed please justify the reasons. Please refer to our statistical reporting guidelines for assistance (https://journals.plos.org/plosone/s/submission-guidelines.#loc-statistical-reporting).

4. In your statistical analyses, please state whether you accounted for clustering by locality. For example, did you consider using multilevel models?

5. We noted in your submission details that a portion of your manuscript may have been presented or published elsewhere. 

"These results have been submitted as an abstract to the 2021 American Heart Association Epidemiology and Lifestyle Meeting. The abstract has not yet been accepted or published."

6. We note you have included a table to which you do not refer in the text of your manuscript. Please ensure that you refer to Table 4 in your text; if accepted, production will need this reference to link the reader to the Table.

Reviewers' comments:

Reviewer's Responses to Questions

**Comments to the Author**

1. Is the manuscript technically sound, and do the data support the conclusions?

Reviewer #1: No

Reviewer #2: Yes

2. Has the statistical analysis been performed appropriately and rigorously? 

Reviewer #1: No

Reviewer #2: No

3. Have the authors made all data underlying the findings in their manuscript fully available?

Reviewer #1: No

Reviewer #2: Yes

4. Is the manuscript presented in an intelligible fashion and written in standard English?

Reviewer #1: Yes

Reviewer #2: Yes

5. Review Comments to the Author

Reviewer #1: Thank you for the invitation to review this manuscript. Overall, this is an interesting research topic but may need substantial revision. I have highlighted several major methodological concerns below that require the authors’ attention. There were also minor comments that did not include here as they are od lesser importance and I have prioritised the major ones.

Major comments

1. Introduction – comprehensive and sufficient

2. Methods

a. Please describe how dietary recall was done in children. Were proxies used?

b. Why was 1 X 24h recall used instead of both recalls? This is an important point esp. the analyses were predominantly based on micronutrients, that vary significantly from day-to-day? Usual intake (instead of actual) intake may be a better reflection of habitual intake, and more accurate representation of diet quality.

c. What was the difference between CFS and BNS food groups? Why was BNS needed if the CFS provides top WG sources too?

d. Whole grain determination should be clearly described – did they include all foods from Tier 1-4 or only those from major food groups but not discretionary foods (e.g. high fat high sugar desserts made with wholemeal flour)? What happened to the 49% food such as mixed dish that can contribute to whole grain intake that were not categorised in the CFG?

e. Whole grain was not seen in the CFG Table S1. Please check.

f. Diet quality – how was NRF chosen. Diet quality should consider intake of food groups and adherence to dietary guidelines, instead of just selected nutrients. Can the authors cite studies that NRF has been validated against other diet quality measures such as the healthy eating index etc.?

g. Has the Nutrient-Rich Food Index 9.3 validated for children, or some adults whose intakes were below 2000 kcal/day? When standardising dietary intake to 2000 kcal, was it a fair assumption that intake of nutrients and whole grain increased or decreased linearly during (upward or downward) energy standardisation?

h. Methods should clearly describe how the ‘moderation’ component scores are being deducted (e.g. only proportion that exceeded recommendation)

3. Analytical methods for children and adults should be considered carefully:

a. Covariates used in analysis – are they all appropriate/relevant for children?

b. Covariates – was there an overlap in adjusting sex, BMI, and EI (e.g. females or lean individuals = lower EI)? Was physical activity level considered in adjusting dietary intake? Was NRF adjusted, if yes, was EI over-adjusted if NRF already considered per 2000 kcal intake?

c. Post-hoc analysis could be performed to show participants which WG categories the significant differences came from.

4. Results

a. Overall, findings (demographics, nutrient intake, diet quality) should be analysed and reported separately for children and adults, and some reasons are (but not limited to) listed above. Also, if combined, young children may be categorised into the low whole grain categories due to their lower total energy intake, hence skewed the results reported in this manuscript.

b. I suggest to report unadjusted nutrient intake, food group, and diet quality intake but include p values for the adjusted stats.

c. Energy intake and whole grain should be included in Table 2 for each WG category according to children and adults.

d. The authors stated that higher WG was seen with higher EI because of higher overall intake. In this case, could the intake of macronutrients be reported as a percentage of total energy intake? Just a thought.

e. Food group results – as above.

5. Discussion

a. Discussion and conclusion need to be revised based on the results after data-re-analysis

b. Overall lacks depth and the authors could interpret/discuss the findings by considering other food groups too. For examples, could the higher fibre intake found in the high WG group explained by other fibre-rich foods e.g. fruits/vegetables/legumes?

c. Several epidemiological studies have examined the relationships between WG and nutrients intake (e.g. NHANES, UK National Diet and Nutrition Survey), but these studies were not mentioned in the discussion of findings

d. I suggest the authors to consider separating discussion from results. For example, the authors reported and discussed nutrient intake, which involves food group and diet quality factors that were not reported until much later in the manuscript.

Reviewer #2: The article presents the findings of the Canadian national nutrition survey examining nutrient intake and diet quality of participants with differing wholegrain intake. A less stringent definition of wholegrains were used than provided in the Canadian Dietary Guidelines. It would be interesting to compare how this has impacted the findings and strength of the conclusions. In addition I have the following comments:

• Were replicate weights and proc surveyreg /proc surveyfreq used for the analysis?

• Misreporting is not described: how many participants misreported energy in the survey? How do the results differ after accounting for misreporting?

6. PLOS authors have the option to publish the peer review history of their article (what does this mean?). If published, this will include your full peer review and any attached files.

Reviewer #1: No

Reviewer #2: No

---

## [Author Response · Author response to Decision Letter 0]

26 Feb 2021

Please see attached "Response to Reviewers" for a point-by-point response to all of the editors' and reviewers' comments.

---

## [Decision Letter · Decision Letter 1]

29 Apr 2021

PONE-D-20-36067R1

Association between whole grain food intake in Canada and nutrient intake, food group intake and diet quality: Findings from the 2015 Canadian Community Health Survey

PLOS ONE

Dear Dr. Smith,

Thank you for submitting your manuscript to PLOS ONE. After careful consideration, we feel that it has merit but does not fully meet PLOS ONE’s publication criteria as it currently stands. Therefore, we invite you to submit a revised version of the manuscript that addresses the points raised during the review process.

We look forward to receiving your revised manuscript.

Kind regards,

Jimmy Louie

Academic Editor

PLOS ONE

Additional Editor Comments (if provided):

Please take care to address the remaining comments of the reviewers.

Reviewers' comments:

Reviewer's Responses to Questions

**Comments to the Author**

1. If the authors have adequately addressed your comments raised in a previous round of review and you feel that this manuscript is now acceptable for publication, you may indicate that here to bypass the “Comments to the Author” section, enter your conflict of interest statement in the “Confidential to Editor” section, and submit your "Accept" recommendation.

Reviewer #1: (No Response)

Reviewer #2: All comments have been addressed

2. Is the manuscript technically sound, and do the data support the conclusions?

Reviewer #1: Yes

Reviewer #2: Partly

3. Has the statistical analysis been performed appropriately and rigorously? 

Reviewer #1: Yes

Reviewer #2: Yes

4. Have the authors made all data underlying the findings in their manuscript fully available?

Reviewer #1: Yes

Reviewer #2: Yes

5. Is the manuscript presented in an intelligible fashion and written in standard English?

Reviewer #1: Yes

Reviewer #2: Yes

6. Review Comments to the Author

Reviewer #1: The authors have addressed most of my comments (thank you), and I have a few remaining or new comments for the authors' consideration:

1. In the Methods Diet Quality section, the authors state that total sugar was used instead of added sugar (line 165). Does this affect the validity of this score? Total sugar includes sugars from fruits and dairy and it does not make sense to deduct points for the consumption of sugar from these sources compared to added sugar?

2. Regarding response to question 3b, have you checked for co-linearity between sex, BMI and EI?

3. I agree that post-hoc comparisons may be too much for nutrients intake, but I suggest the author to do this for the NRF scores. This relates to Point 6 below.

4. Table 2 - Title says 'adjusted' but EI is 'Unadjusted'

5. Table 3 (applies to supplement) - can you add a comma before the unit, and should it be g/day?

6. Discussion line 376-377 - the statement suggests a dose-response relationship, but the diet quality scores do not appear to be different between low- and mid- intake groups. Please add post-hoc comparisons to support this statement if this was indeed the case.

7. Discussion line 406 - Please clarify what 'complementary' means in term of the prevalence of obesity in this study and Hosseini et al.

8. Discussion line 421 - You cited Ref 37 Hosseini but I cannot see diet quality being reported in the cited study?

9. Discussion Line 438 - A previous study also analysed the sources of whole grains in children and adults, see Galea et al, Public Health Nutrition 2017;20(12)"2166-2172.

10. Line 458 - in this study there was no significant difference in overweight and obesity prevalence between groups (Table 1) so I think this statement could be removed.

Reviewer #2: 1.Many dietary guidelines recommend that wholegrains compose 50% of grain foods consumed. Can the authors please provide the proportion of grain foods that are wholegrain?

2.The dataset is not disaggregated. How were the amounts of wholegrains in mixed dishes assessed?

3.Please conduct a sensitivity analysis to determine the effect of energy misreporting on the analysis and report the proportion of plausible reporters in Table 1. Please use a method to determine misreporting that has been validated for 1 day of dietary data and cite the paper that validates the method not just previous research that has used the method. Two validated methods include the Goldberg equation or energy prediction equations reported by Huang, 2005, Obesity Res.

4.It would be of greater value to the literature to know which subgroups of the population have a diet composition composed of less wholegrain relative to requirements. This will better reflect which groups should increase intake to meet recommended amounts of wholegrains. I suggest categorizing participants as grams per 1000 kcal as suggested: https://dietassessmentprimer.cancer.gov/learn/adjustment.html.

5. It is a limitation that grams rather than serves are used to assess wholegrain intake. Is there a reason why serves couldn't be assessed?

6. It should be argued that participants that consumed more wholegrains also consumed more of other nutrient-dense foods and therefore increasing wholegrains is not necessarily an independent component to contribute to better diet quality based on the present analysis. The conclusion should be altered accordingly

7. PLOS authors have the option to publish the peer review history of their article (what does this mean?). If published, this will include your full peer review and any attached files.

Reviewer #1: No

Reviewer #2: No

---

## [Author Response · Author response to Decision Letter 1]

14 May 2021

Please see attached document with detailed responses to each of the reviewer's comments.

---

## [Editor Report · Decision Letter 2]

28 May 2021

Association between whole grain food intake in Canada and nutrient intake, food group intake and diet quality: Findings from the 2015 Canadian Community Health Survey

PONE-D-20-36067R2

Dear Dr. Smith,

We’re pleased to inform you that your manuscript has been judged scientifically suitable for publication and will be formally accepted for publication once it meets all outstanding technical requirements.

Kind regards,

Jimmy Louie

Academic Editor

PLOS ONE
---

## [Editor Report · Acceptance letter]

22 Jun 2021

PONE-D-20-36067R2 

 Association between whole grain food intake in Canada and nutrient intake, food group intake and diet quality: Findings from the 2015 Canadian Community Health Survey 

Dear Dr. Smith:

I'm pleased to inform you that your manuscript has been deemed suitable for publication in PLOS ONE. Congratulations! Your manuscript is now with our production department. 

Kind regards, 

on behalf of

Dr. Jimmy Louie 

Academic Editor

PLOS ONE